# OpenReview forum: "The Silent Judge: Unacknowledged Shortcut Bias in LLM-as-a-Judge"
_NeurIPS.cc/2025/Workshop/Reliable_ML — NeurIPS 2025 - Reliable ML Workshop_

### Official Review · Reviewer_PKFu · 2025-09-19
**Review: The Silent Judge**

**Rating:** 6
**Confidence:** 3

**Review:**

This paper addresses unacknowledged shortcut biases in LLM-as-a-Judge evaluation protocols. The premise is that LLMs may rely on superficial cues rather than the content itself when judging responses. Furthermore, they fail to acknowledge this behavior in their justifications. The authors test this hypothesis by introducing cues to two datasets, ELI5 and LitBench, and evaluating the responses of GPT-4o and Gemini-2.5-Flash. The findings of a strong recency bias and a provenance hierarchy (EXPERT > HUMAN > LLM > UNKNOWN) are compelling. Moreover, it is interesting that the justifications rationalise the decisions with content-based arguments while ignoring the cues.

However, the paper has several limitations. First, the analysis is largely descriptive demonstrating that the biases exist. However, there is no   discussion on why they exist. For example, is the recency bias a learned behaviour from training data that favours new information? Is the provenance hierarchy a reflection of the models internal "trust" scores for different data sources? Second, the study includes two datasets. This provides a strong starting point, however, it raises questions about the generalisability of the findings. The paper would be stronger if it included a broader range of LLMs (e.g., open-source models) and more diverse evaluation tasks.

In conclusion this is a novel finding regarding LLM justification failures. The research question is relevant and the methodology is sound. I recommend a weak accept.

---

### Official Review · Reviewer_qifQ · 2025-09-21

**Rating:** 7
**Confidence:** 3

**Review:**

### Summary

This work studies the reliability of using LLMs as judges to evaluate system outputs. The authors test whether inserting *superficial cues* (e.g., “written by HUMAN”) into the prompt affects the judge’s verdict. They also check whether the LLM *acknowledges* these cues in its rationale.

Experiments are run on **GPT-4o** and **Gemini-2.5-Flash** using two datasets of 100 evaluation tasks each. In each task, the LLM must choose between two responses to the same prompt; each response is labeled by **provenance/expertise** (Expert, Human, LLM, Unknown) and by **recency** (Old, New).

The authors find that LLMs show clear bias under both expertise and recency cues. Moreover, the LLMs do not acknowledge these cues in their rationales and instead provide generic reasons for their choices.

---

### Strengths

- The paper is well written; the setting and results are presented clearly.
- Decoding uses temperature = 0 (greedy) with a fixed seed, making outcomes reproducible.
- The question is important: LLMs are widely used as judges in practice.

---

### Weaknesses

- Narrow scope: only two models and two datasets, with 100 items per dataset. This limits how strongly the findings generalize.
- No uncertainty quantification: results are reported without confidence intervals or significance tests, despite the small sample size.
- On LitBench dataset, the “Expert” label is not included, yet the paper discusses a provenance preference order (Expert > Human > LLM > Unknown). This weakens that specific claim.

---

### Suggestions

- Expand the experiments: more models and datasets to strengthen the results.
- Report uncertainty: add confidence intervals for verdict-shift rates and acknowledgment rates (the two metrics used by the paper).
- Test cue robustness: vary cue wording, placement (before/after responses), and formatting to see if the effects persist.
- Consider subtler cues (e.g., light phrasing embedded inside the text) to model realistic attempts to influence the judge.

---

### Overall

I believe the paper studies an urgent problem and provides important preliminary findings. However, for the results to be more convincing and imposing, further experiments should be contacted.